# Body Appreciation, Weight Status, and Weight Management Practices Among First-Year Students at Universities of Applied Sciences in Lithuania

**DOI:** 10.3390/medicina61071223

**Published:** 2025-07-05

**Authors:** Vilma Kriaučionienė, Asta Raskilienė, Lina Šnipaitienė, Janina Petkevičienė

**Affiliations:** 1Health Research Institute, Faculty of Public Health, Lithuanian University of Health Sciences, Tilžės Str. 18, 47181 Kaunas, Lithuania; asta.raskiliene@lsmu.lt (A.R.); janina.petkeviciene@lsmu.lt (J.P.); 2Department of Preventive Medicine, Faculty of Public Health, Lithuanian University of Health Sciences, Tilžės Str. 18, 47181 Kaunas, Lithuania; lina.snipaitiene@lsmu.lt; 3Institute of Biological Systems and Genetic Research, Lithuanian University of Health Sciences, Eivenių Str. 4, 44307 Kaunas, Lithuania

**Keywords:** body appreciation, body weight status, weight perception, weight management practices, university students

## Abstract

*Background and Objectives:* The associations between body appreciation, weight status, and weight management practices are influenced by societal, cultural, and psychological factors. Studies indicated that a higher level of body appreciation is linked to lower engagement in unhealthy weight management practices. The transition from high school to university is a significant life event, often accompanied by substantial lifestyle changes that can affect students’ body image and weight-related behaviours. This study aimed to assess the associations between body appreciation, weight status, and weight management behaviours among first-year students at four universities of applied sciences in Lithuania. *Materials and Methods*: A cross-sectional online survey was conducted in 2022 among 709 first-year students (216 males and 493 females) at the four largest universities of applied sciences in Vilnius, Kaunas, Klaipėda, and Šiauliai. Body appreciation was assessed using the Body Appreciation Scale-2 (BAS-2), whilst BMI was calculated from self-reported height and weight. Students were asked about their weight management practices. Logistic regression analysis was applied to evaluate associations between weight management behaviours, body appreciation, and weight status. *Results:* Female students had a significantly lower median BMI (21.1 kg/m^2^) than males (23.3 kg/m^2^) but were more likely to perceive themselves as overweight (34.5% vs. 17.1%), worry about gaining weight (40.6% vs. 11.6%), and attempt weight loss (52.5% vs. 23.6%) (all *p* < 0.001). Higher BAS scores were associated with greater accuracy in weight perception, higher satisfaction with body weight, and fewer concerns about weight gain. Students with lower BAS scores were more likely to engage in harmful weight-control behaviours such as smoking (OR = 0.07; 95% CI: 0.02–0.25 for high vs. low BAS) and were more strongly influenced by media beauty standards and dissatisfaction with appearance. *Conclusions:* Body appreciation is linked to healthier weight perceptions and behaviours. Interventions that enhance body appreciation may help reduce body dissatisfaction and prevent unhealthy weight control practices, especially among female students.

## 1. Introduction

Body image is a complex concept that includes how individuals perceive, think about, and feel emotionally towards their bodies. This concept plays an important role in psychological well-being and behaviour, particularly among young people [1,2]. One of the key indicators of a positive body image is body appreciation, which is the ability to accept and respect one’s own body, reflecting sociocultural ideals of appearance and unrealistic beauty standards [3]. Research shows that body appreciation is positively associated with higher self-esteem, psychological well-being, and a greater engagement in health-promoting behaviours [4,5].

The Body Appreciation Scale-2 (BAS-2) was developed and validated to measure body appraisal across various populations. The BAS-2 effectively reflects body acceptance and resilience against negative societal influences [1,3,6]. It is widely used in diverse cultural contexts and is sensitive to differences in gender and age [6].

The transition from secondary school to university is a crucial time in a young person’s development. During this period, students typically leave the parental home, gain more independence in decision-making, and face new academic and social challenges. These changes can impact their lifestyle habits and attitudes towards body weight and appearance [7]. Increased exposure to peer norms and media representations of idealized body shapes may contribute to body dissatisfaction among university students [8,9].

Previous studies have identified significant differences in body image and related behaviours between the sexes. Female students are more likely to perceive themselves as overweight, express dissatisfaction with their appearance, and attempt to lose weight even when their BMI falls within the normal range [1,7,10,11]. In contrast, male students often desire a more muscular body and may underestimate their body weight or body fatness, focusing more on muscularity than leanness [12,13]. These different ideals contribute to gender-specific patterns of body image and health behaviour [5,14].

Body weight is a significant factor that affects body assessment. A higher BMI is often linked to lower body satisfaction and body appreciation among adolescents and young adults [15,16,17]. However, individuals with a higher body weight can still maintain a sense of high body appreciation if they reject negative societal messages and develop a more accepting attitude towards their bodies [11]. Furthermore, research indicates that perceived body weight does not always match actual BMI. Inaccurate self-perception, particularly among females, can lead to inappropriate or harmful weight control behaviours such as smoking, extreme dieting, or skipping meals [17,18].

While the issue of body image has gained increasing attention globally, research on body appreciation and weight perception in Eastern European Union countries is still limited. In Lithuania, young adults have been raised in a rapidly changing socio-cultural environment, which includes heightened exposure to Western ideals. Therefore, it is important to investigate how these factors influence students’ body image- and weight-related behaviours.

The aim of this study was to evaluate the associations between body appreciation, actual and perceived body weight status, and weight-control practices among first-year university students in Lithuania. Understanding these associations is crucial for developing health promotion strategies that encourage realistic body perceptions and support healthy lifestyle choices among university students.

## 2. Materials and Methods

### 2.1. Study Design and Sample

This study used an online cross-sectional design involving first-year students from the largest universities of applied sciences in Lithuania. The participating institutions were located in the following four major cities: Vilnius, Kaunas, Klaipėda, and Šiauliai. Data collection took place during the second semester of the 2021–2022 academic year. Faculties were randomly selected within each institution, and all enrolled first-year students in these faculties were invited to participate via institutional email. No exclusion criteria were applied. The emails included a description of the survey and a link to the questionnaire. The survey was available for three weeks, during which reminder emails were sent in the first and second weeks to enhance response rates. Participation was completely voluntary and anonymous, with a strong emphasis on confidentiality to encourage truthful responses. Out of the 3253 students invited to participate in the study, 721 (221 males and 500 females) completed the self-administered questionnaire (response rate 22.2%). Ethical approval for the study was obtained from the Bioethics Centre of the Lithuanian University of Health Sciences (approval numbers BEC-GVM(M)-80 on 28 February 2022 and BEC-GM(M)-119 on 13 April 2021). Institutional approval was also obtained from the university administration to ensure compliance with local research governance standards.

### 2.2. Measurements

A self-developed questionnaire was created for this study to evaluate actual and perceived weight status, body weight satisfaction, concerns about weight gain, efforts to lose weight, and factors that encourage weight loss. The questions about weight perception, satisfaction with weight, and concerns about weight gain were selected from validated questionnaires used in our prior student research [19]. Respondents provided self-reported measurements of height and weight. Body mass index (BMI) was calculated by dividing reported weight (in kilograms) by height squared (in metres) and was categorized according to the following World Health Organization guidelines: underweight (<18.5 kg/m^2^), normal weight (18.5–24.9 kg/m^2^), overweight (25.0–29.9 kg/m^2^), and obese (≥30 kg/m^2^). To assess perceived weight status, participants were asked how they viewed their body size using five response categories: ‘much too thin’, ‘a little too thin’, ‘about right’, ‘a little overweight’, and ‘very overweight’. For analysis, these responses were recoded into the following three groups: (1) ‘too thin’, (2) ‘just right’, and (3) ‘too fat’. The accuracy of weight perception was determined by comparing participants’ BMI classification with their self-perceived weight category, categorizing them as having either accurate or inaccurate weight perception. Body weight satisfaction was measured with the question, “How satisfied are you with your weight?” Participants had four response options ranging from ‘very satisfied’ to ‘very dissatisfied’. Responses were combined into the following two groups: satisfied (‘very satisfied’ and ‘somewhat satisfied’) and dissatisfied (‘somewhat dissatisfied’ or ‘very dissatisfied’). Concerns about weight gain were assessed by asking whether students worried about gaining weight, with response options grouped as ‘worried’ (responses: ‘very often’ and ‘often’) and ‘not worried’ (responses: ‘never’, ‘rarely’, or ‘sometimes’). Efforts to lose weight in the past 12 months were also examined. Participants were asked if they had attempted to lose weight and, if so, which methods they had used. The response options included consuming low-fat foods, counting calories, reducing portion sizes, following specific weight-loss diets, engaging in vigorous exercise, using extreme weight-loss strategies (such as laxatives, pills, or weight-loss supplements), and smoking. Students had the option to select multiple weight-loss methods.

To identify factors that encourage weight loss, students were asked “What do you think are the reasons for dieting to regulate your body weight?” Eight possible answers were provided, including ‘dissatisfaction with appearance, lack of confidence’, ‘the belief that having a slender body makes it easier to achieve desired goals’, ‘comments from people around about an imperfect body’, ‘negative societal attitudes towards people with overweight’, ‘media shaping beauty standards’, ‘desire to try new diets’, ‘health problems’, and ‘other’.

The Body Appreciation Scale-2 (BAS-2) was used to evaluate students’ attitudes towards their bodies. This 10-item scale uses a 5-point Likert format, where 1 means ‘never’ and 5 means ‘always’. The total scores range from 10 to 50, with higher scores indicating a more positive body image. The BAS has been previously validated among students in Lithuania, demonstrating that it is a valid tool for measuring body appreciation [20]. The Lithuanian version of the scale demonstrated strong reliability (Cronbach’s alpha = 0.962).

### 2.3. Statistical Analysis

All analyses were conducted using IBM SPSS Statistics for Windows, version 20.0 (IBM Corp., Armonk, NY, USA). Comparisons of categorical data were performed using Chi-squared tests and post hoc z-tests with Bonferroni adjustments for multiple comparisons. Since the assumptions of normal distribution were not met (as assessed by the Kolmogorov–Smirnov test), continuous variables were presented as medians along with interquartile ranges. *p* values of less than 0.05 were considered to be statistically significant.

To evaluate the internal consistency of the BAS-2 scale, we calculated Cronbach’s alpha, which was found to be 0.962. As most of the analyzed variables were categorical, we divided the BAS-2 scores into tertiles using the cutoff points of 29 and 37. The tertiles represent low, medium, and high levels of body appreciation, allowing us to apply statistical analyses suitable for categorical variables.

A multivariable logistic regression analysis was conducted to investigate the associations between weight reduction practices and factors such as gender, body appreciation, and BMI. Additionally, this analysis was used to evaluate how the factors that promote weight loss are associated with gender, body appreciation, and BMI.

## 3. Results

The main characteristics of the first-year university student population are presented in Table 1. The median age of male participants was 19 years, while for females it was 20 years. There was no significant difference between the genders in terms of age (*p* = 0.263). The median score of the BAS was 34.0 for males and 33.0 for females, which also showed no significant difference (*p* = 0.279). However, males had a significantly higher median BMI of 23.3 kg/m^2^ compared to females, who had a median BMI of 21.1 kg/m^2^ (*p* = 0.001).

The perception of body weight differed significantly between males and females (*p* < 0.001). Males were more likely to view themselves as too thin, while a higher proportion of females considered themselves to be overweight. Females demonstrated a higher accuracy in weight perception compared to males. Overall satisfaction with body weight was similar for both genders, with 56.0% of males and females reporting satisfaction. However, concerns about gaining weight were significantly more common in females (40.6%) than in males (11.6%) (*p* < 0.001). Additionally, a significantly greater proportion of females (52.5%) attempted to lose weight compared to males (23.6%) (*p* < 0.001) (Table 1).

Among students categorized as either overweight or normal weight, perceptions of weight varied significantly across BAS score tertiles (*p* < 0.001). Students with the lowest BAS scores were most likely to perceive themselves as overweight (Table 2). However, weight perception accuracy was highest among students with a higher BMI and a lower BAS score. Conversely, within the normal body weight group, weight perception accuracy was highest among those with the highest BAS scores.

Among students with a normal BMI, satisfaction with body weight was significantly higher among individuals in the highest BAS tertile, with 73.4% reporting satisfaction, compared to only 47.1% in the lowest BAS tertile (*p* < 0.001). Concerns about gaining weight also varied significantly across BAS tertiles. Among individuals in the lowest BAS tertile, 44.1% expressed worry about gaining weight, while this concern decreased to just 10.1% in the highest BAS tertile among those with a BMI under 25. In contrast, concerns were significantly higher among students with overweight, with 81.3% in the lowest BAS tertile and 21.6% in the highest BAS tertile (*p* < 0.001).

Students with a normal BMI and the lowest BAS scores were more likely to attempt weight loss. In contrast, a significant proportion of students classified as overweight tried to lose weight across all BAS tertiles, although the prevalence of attempts decreased in the higher BAS groups. Furthermore, the differences in success rates for weight loss efforts between these groups were significant. This suggests that students with lower BAS scores face more challenges in achieving successful weight reduction.

Data from a multivariable logistic regression analysis indicated that female students were significantly more likely than male students to engage in weight reduction practices, which included using low-fat food products, counting calories, reducing food intake, following a special diet, and smoking (Table 3). Additionally, students who were classified as overweight were significantly more likely to use all of these methods, except smoking, compared to those with a lower BMI (*p* < 0.01 for all variables).

Higher scores of the BAS were inversely associated with specific weight reduction behaviours. Students with the highest BAS scores were significantly less likely to use low-fat food products (OR = 0.52; CI: 0.33–0.84; *p* = 0.007) and less likely to smoke (OR = 0.07; CI: 0.02–0.25; *p* < 0.001) compared to those with the lowest BAS scores. However, BAS score did not significantly predict engagement in intensive exercise or other dietary modification techniques (Table 3).

Figure 1 illustrates the proportion of students who identified some motivating factors for weight loss. The main motivators included dissatisfaction with appearance, lack of confidence, societal attitudes toward obesity, and media-driven beauty standards.

Dissatisfaction with appearance and lack of confidence were significantly stronger motivators for weight loss among females, with 81.5% of women reporting these feelings compared to 74.9% of men. Additionally, beauty standards shaped by the media influenced more female students (40.2%) than male students (31.6%), and comments from people about imperfect bodies affected more females (50.8%) than males (41.4%). This highlights the greater societal pressure on females regarding body image.

Results from the multivariable logistic regression analyses showed that female students were 1.42 times more likely to report dissatisfaction with appearance as a motivator for weight loss compared to male students. Additionally, media-driven beauty standards influenced weight loss motivation for females more than for males (Table 4).

Students with lower BAS scores were significantly more likely to be motivated to lose weight due to dissatisfaction with appearance, lack of confidence, and negative public attitudes towards obesity compared to those with the highest BAS scores. The influence of beauty standards was also more pronounced among students with lower BAS scores.

Students with overweight were significantly less likely to report negative public attitudes toward people with obesity as a motivator for weight loss compared to those with a BMI of less than 25 kg/m^2^ (OR = 0.49; CI: 0.33–0.70; *p* < 0.001). Similarly, students with a higher BMI were less likely to mention beauty standards as a motivator for weight loss (OR = 0.64; CI: 0.44–0.91; *p* = 0.015) (Table 4).

## 4. Discussion

This study examined body appreciation and its associations with BMI, body weight perception, and weight control behaviours among first-year students at the four biggest universities of applied sciences in Lithuania. Our findings are consistent with previous studies, demonstrating that body image is influenced by a complex interplay of anthropometric characteristics, sociocultural norms, and media influence, particularly during significant life transitions such as starting university [21].

During their first year of university, students undergo various psychological and behavioural changes related to increased autonomy, identity development, academic responsibilities, and social interactions [22,23]. These transitional experiences can impact students’ body weight and health-related behaviours. Internal conflicts between their actual appearance and perceived societal ideals, often influenced by the media and peers, may lead to lower self-esteem and increased anxiety.

Our findings support prior evidence that female students reported significantly higher levels of body dissatisfaction and greater concerns about body shape compared to male students, despite having a lower BMI [24,25]. This aligns with previous studies that highlight thinness as a cultural ideal for women, while muscularity is often viewed as desirable for men [1,22,26]. Women often perceive themselves as overweight, regardless of their actual BMI. They are more likely to engage in a variety of weight reduction practices, such as using low-fat food products, counting calories, reducing food intake, fasting, following specific diets, eating food substitutes, and even smoking— behaviours increasingly prevalent among young women [22,26,27,28,29]. Males generally reported greater body appreciation. However, there has been an increase in dissatisfaction related to muscularity and body size among males, likely influenced by idealized masculine images in social and digital media [22,30]. Male students who are underweight may feel they do not meet societal standards of strength and masculinity, which can lead to a negative body image and psychological distress [30,31].

Students with higher body appreciation in our study were less likely to perceive themselves as overweight and more likely to express satisfaction with their bodies. Notably, they reported fewer weight loss behaviours, even when overweight. When weight was managed, healthier approaches were adopted and smoking was less common. These findings support prior research suggesting that body appreciation can serve as a protective factor against dissatisfaction and disordered eating [2,4,31].

This phenomenon may indicate a deeper psychological mechanism. We suggest that students with higher body appreciation are more likely to prioritize their overall well-being and body functionality rather than focusing solely on appearance. This approach may help to reduce the emphasis on appearance-based weight control. Additionally, these individuals are more likely to reject societal beauty ideals and show a lower susceptibility to body-related stigma [20,32]. Additionally, recent research emphasizes the importance of self-esteem in shaping body image among university students, especially during early adulthood when concerns about social status and appearance are more pronounced. Findings indicate that lower self-esteem is associated with a higher motivation to change one’s weight, particularly among females [33]. Improving self-esteem has been shown to foster body appreciation and decrease vulnerability to internalized stigma. These findings may help explain why students who have a positive appreciation of their bodies tend to report greater satisfaction with their weight and are less likely to engage in restrictive weight management behaviours, even if their BMI exceeds normal thresholds.

The misperception of body weight status is linked to lower body appreciation. Students with a normal BMI who considered themselves as overweight tend to have significantly lower BAS-2 scores and were more likely to attempt weight loss. These findings are in line with previous research indicating that many young adults misperceive their body weight, which can lead to psychological stress, disordered eating, and low self-esteem [17,18,22]. Both male and female students with distorted perceptions of their weight reported more unhealthy weight-control behaviours [32], and such distortions have been found to be associated with sadness and suicidal thoughts [27].

Higher body appreciation may encourage students to prioritize health and body functionality over appearance, thus decreasing the perceived need for weight control. Our research aligns with other studies in Lithuania showing that higher BAS scores are linked to increased self-esteem and emotional resilience, which can help to reduce internalized stigma and distress related to appearance [34].

The impact of social media on body image has been widely recognized. Platforms like Instagram and TikTok promote unrealistic beauty standards and foster social comparison, especially among females [35,36]. These effects are particularly pronounced among users who engage heavily with these platforms and those who are more susceptible to comparison. Research shows that visual platforms are more detrimental than text-based ones. Media literacy and emotional well-being can serve as protective factors against these negative effects [37].

Our data showed that students with overweight were less likely to report negative public attitudes towards obesity and beauty standards as motivating factors for weight control. Obesity stigma remains a significant issue that can impact both behaviour and emotional well-being. Individuals with a higher body weight often face negative stereotypes, such as assumptions of laziness or a lack of self-control [38]. This stigmatization is reinforced by media portrayals and societal discourse, leading to internalized shame, low self-esteem, and the avoidance of health-promoting behaviours [39,40,41]. The language used to describe body size can significantly impact people’s attitudes. Research indicates that using weight-neutral and fat-related terms tends to carry less stigma and encourages more supportive health behaviours, particularly for individuals with larger bodies, compared to medicalized language [42]. Internalized stigma contributes to shame, avoidance of health-promoting behaviours, and psychological distress. In university settings, stigma can lead to social withdrawal and may create barriers to seeking help and engaging in healthy weight management.

Our findings are consistent with national data indicating that Lithuanian female students have a more accurate perception of their weight status compared to males, but they express greater dissatisfaction [20]. Longitudinal data from studies conducted among university students in Kaunas between 2000 and 2017 indicated that perceptions of body image and weight control behaviours remained stable over two decades. Notably, it was subjective perceptions, rather than objective weight status, that influenced weight control behaviours [23]. This paradox, where a more accurate perception of body image aligns with greater dissatisfaction, has not been extensively documented in Lithuanian literature. It may suggest that young Lithuanian women are internalizing Western ideals of thinness.

Our results highlight gender-specific patterns and sociocultural influences. Female students not only reported more dissatisfaction and weight-related behaviours, but also greater media-driven pressure to conform to thin ideals. In contrast, males were more likely to perceive themselves as too thin and showed less concern about weight gain. These patterns, also seen in previous Lithuanian studies, suggest that Western beauty norms are integrated into local contexts shaped by societal transformation [43].

To address body image concerns among university students, it is crucial to promote accurate weight perception, strengthen media literacy, and provide access to gender-sensitive psychological support. Body positivity campaigns and media literacy interventions have been shown to reduce the internalization of harmful beauty ideals and improve body image outcomes [9,37]. In Lithuania, most universities have already taken practical steps to address student well-being. For example, many institutions offer free, anonymous psychological consultations to address stress and mental health concerns. A unique example is the Lithuanian University of Health Sciences, which provides lifestyle medicine consultations for students focused on individualized strategies in physical activity, nutrition, stress management, and sleep. These services, offered free of charge, help students develop healthier lifestyle habits that can support a more positive body image and reduce reliance on appearance-based weight control. Integrating such services into broader university health promotion strategies may serve as a valuable model for enhancing both physical and emotional well-being in Lithuanian and international academic settings.

This study provides important insights into the associations between body appreciation, BMI, body weight perception, and related weight management behaviours among university students. It enhances existing Lithuanian research by examining the link between weight regulation techniques and body appreciation, an area that has received little attention. However, several limitations should be considered when interpreting the findings. The primary limitation of this study is its cross-sectional design, which does not allow for the establishment of causal relationships. The study relies on self-reported data for height and weight. Previous research has shown that self-reported anthropometric measurements are generally accurate for estimating BMI at the population level; however, there is still a risk of reporting bias [44]. Participants may underreport their weight or overestimate their height. University students might provide more accurate reports due to their higher education status and young age. In Lithuania, it is mandatory to take anthropometric measurements for every school child before the start of the school year. Thus, most young Lithuanians are aware of their height and weight. Additionally, our study was anonymous, which ensured confidentiality and encouraged participants to report accurate data. While most studies have found limited underestimation of self-reported BMI, which can affect prevalence estimates for BMI categories, the researchers agree that using self-reported BMI is appropriate for association analyses. Social desirability bias may also influence responses to other questions, for example, leading to the underreporting of unhealthy weight control behaviours. Selection bias may be a concern since participation was voluntary, which could have attracted students particularly interested in health or body image issues. To enhance sample diversity, we included students from different faculties across four major universities of applied sciences and a variety of study programmes. The overall response rate was modest, which is a common limitation of online surveys targeting students and limited the generalizability of the study results. First-year students may differ from older students, which also limits the study results generalizability. Additionally, male participation was significantly lower, partly due to the gender distribution of students within the selected study programmes. This discrepancy may affect the reliability of gender-based comparisons. Weight control methods listed in the questionnaire do not include all the techniques that students might use. Important data on several potentially relevant factors, such as socioeconomic status, psychological well-being, and media usage patterns, were not collected. These contextual factors could influence body image and weight management behaviours, as well as the analyzed associations. Future research should take these factors into consideration.

Despite these limitations, the study has several strengths. A large and diverse sample of students from different universities across the country and from different academic programmes was included, which increases the representativeness of the data. The use of the validated Body Appreciation Scale-2 (BAS-2) provided reliable measurements of positive body image, enabling meaningful comparisons across different subgroups.

Importantly, the study adopted a multidimensional approach, examining body image in the context of BMI classification, weight perception accuracy, and a wide range of weight management-related practices. This allowed for a more comprehensive understanding of the factors contributing to body appreciation in early adulthood and may help to identify targets for future health promotion and education strategies among university students.

## 5. Conclusions

This study demonstrated that body appreciation is associated with more accurate weight perception, greater satisfaction with body weight, and healthier weight management behaviours among university students. Strengthening body appreciation, particularly among female students, may help to reduce dissatisfaction with body weight and prevent engagement in harmful weight control practices. University health promotion programmes should incorporate strategies aimed at enhancing body appreciation, promoting media literacy, providing psychological support, and encouraging healthy lifestyle habits to support realistic body image and health-oriented behaviours during the transition to adulthood.

Future studies should incorporate longitudinal designs and consider the role of variables such as psychological well-being, media exposure patterns, and socioeconomic factors to provide a more comprehensive understanding of the dynamics of body image development in early adulthood.

## Figures and Tables

**Figure 1 medicina-61-01223-f001:**
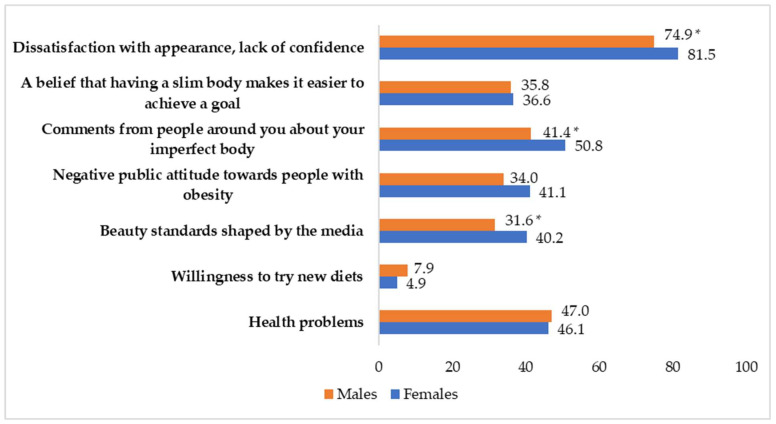
Proportion (%) of students who reported factors that encourage weight loss; *—*p* < 0.05 compared to females.

**Table 1 medicina-61-01223-t001:** Characteristics of the study population.

Characteristics	Male *n* = 216	Female *n* = 493	*p*-Value
**Age** (years), median (IR)	19 (2)	20 (4)	0.263
**Body Appreciation Scale** (scores), median (IR)	34 (10.8)	33 (12.5)	0.279
**Body Appreciation groups** (tertiles) N (%)			0.092
≤29	54 (25.0)	162 (32.9)	
30–37	85 (39.4)	164 (33.3)	
≥38	77 (35.6)	167 (33.9)	
**Body mass index** (kg/m^2^), median (IR)	23.3 (4.6)	21.1 (4.1)	0.001
**Body mass index groups** N (%)			0.009
≤18.5 kg/m^2^	10 (4.6)	52 (10.5)	
18.5–24.99 kg/m^2^	136 (63.0)	319 (64.7)	
≥25 kg/m^2^	70 (32.4)	122 (24.8)	
**Perceived weight status**			<0.001
Too thin	65 (30.1)	47 (9.5)	
Just right	114 (52.8)	276 (56.0)	
Overweight	37 (17.1)	170 (34.5)	
**Accuracy of weight perception**			<0.001
Correct	111 (51.4)	382 (77.5)	
Incorrect	105 (48.6)	111 (22.5)	
**Satisfaction with body weight**			0.993
Satisfied	121 (56.0)	276 (56.0)	
Dissatisfied	95 (44.0)	217 (44.0)	
**Worries about gaining weight**			<0.001
Worried	25 (11.6)	200 (40.6)	
Not worried	191 (88.4)	293 (59.4)	
**Trying to lose weight**			<0.001
Yes	51 (23.6)	259 (52.5)	
No	165 (76.4)	234 (47.5)	

IR—interquartile range; *p*-value from Mann–Whitney or Pearson Chi-square tests.

**Table 2 medicina-61-01223-t002:** Distribution (%) of students by weight perception, body weight satisfaction, concerns about weight gain, and weight reduction efforts according to body appreciation and body mass index.

Variables	BMI (<25)				BMI (≥25)			
	BAS Groups (Tertiles)				BAS Groups (Tertiles)			
	I (≤29)	II (30–37)	III (≥38)	*p*-Value	I (≤29)	II (30–37)	III (≥38)	*p*-Value
**Perceived weight status**								
Too thin	23.5	21.3	19.3	<0.001	0.0	4.0	0.0	<0.001
Just right	47.8 ^a^	66.7	75.4		5.0 ^a^	38.7	54.1	
Overweight	28.7 ^a^	12.1	5.3		95.0 ^a^	57.3	45.9	
**Accuracy of weight perception**								
Correct	54.4 ^a^	70.7	77.3	<0.001	95.0 ^a^	57.3	45.9	<0.001
Incorrect	45.6 ^a^	29.3	22.7		5.0 ^a^	42.7	54.1	
**Satisfaction with body weight**								
Satisfied	47.1	46.0	73.4 ^b^	<0.001	45.0	57.3	59.5	0.199
Dissatisfied	52.9	54.0	26.6 ^b^		55.0	42.7	40.5	
**Worries about gaining weight**								
Worried	44.1 ^a^	28.7 ^c^	10.1	<0.001	81.3 ^a^	28.0	21.6	<0.001
Not worried	55.9 ^a^	71.3 ^c^	89.9		18.8 ^a^	72.0	78.4	
**Trying to lose weight**								
Yes, successfully	23.5	20.1	26.6	<0.001	37.5	37.3	32.4	<0.001
Yes, not successfully	19.1	16.1	4.3 ^b^		43.8 ^a^	17.3	18.9	
No	57.4	63.8	69.1		18.8 ^a^	45.3	48.6	

^a^ *p* < 0.05 compared with BAS II and III groups; ^b^ *p* < 0.05 compared with BAS I and II groups; ^c^ compared with BAS III group—interquartile range; *p*-value from Mann–Whitney or Pearson Chi-square tests.

**Table 3 medicina-61-01223-t003:** Odds ratios (95% CI) of weight reduction practices according to body appreciation and body mass index.

Variables	Weight Reduction Practices					
	Used Low-Fat Food Products		Counted Calories		Moderately Reduced Food Intake	
	OR (95% CI)	*p*-Value	OR (95% CI)	*p*-Value	OR (95% CI)	*p*-Value
**Gender**						
Females vs. males	2.51 (1.62–3.89)	<0.001	2.75 (1.61–4.68)	<0.001	4.00 (2.60–6.16)	<0.001
**BAS**						
30–37 vs. ≤29	0.97 (0.64–1.48)	0.889	0.83 (0.50–1.38)	0.476	0.95 (0.63–1.43)	0.791
≥38 vs. ≤29	0.52 (0.33–0.84)	0.007	1.04 (0.62–1.75)	0.874	0.85 (0.55–1.31)	0.459
**BMI**						
≥25 vs. <25 kg/m^2^	2.33 (1.58–3.42)	<0.001	2.46 (1.58–3.83)	<0.001	2.87 (1.96–4.19)	<0.001
	**Exercised Intensively**		**Applied Special Diets**		**Smoked**	
	**OR (95% CI)**	** *p* ** **-Value**	**OR (95% CI)**	** *p* ** **-Value**	**OR (95% CI)**	** *p* ** **-Value**
**Gender**						
Females	1.10 (0.72–1.65)	0.668	2.43 (1.31–4.49)	0.005	3.83 (1.60–9.16)	0.003
**BAS**						
30–37 vs. ≤29	1.04 (0.65–1.66)	0.880	0.81 (0.47–1.42)	0.468	0.48 (0.26–0.88)	0.017
≥38 vs. ≤29	1.26 (0.78–2.03)	0.349	0.64 (0.34–1.19)	0.160	0.07 (0.02–0.25)	<0.001
**KMI**						
≥25 vs. <25 kg/m^2^	1.68 (1.11–2.54)	0.014	2.85 (1.73–4.68)	<0.001	1.40 (0.78–2.52)	0.263

Abbreviations: BAS—body appreciation scale, BMI—body mass index, OR—odds ratio, CI—confidence intervals.

**Table 4 medicina-61-01223-t004:** Odds ratios for factors that encourage weight loss according to body appreciation and body mass index.

Variables	Factors That Encourage Weight Loss					
	Dissatisfaction with Appearance, Lack of Confidence		Negative Public Attitude Towards People with Obesity		Beauty Standards Shaped by the Media	
	OR (95% CI)	*p*-Value	OR (95% CI)	*p*-Value	OR (95% CI)	*p*-Value
**Gender**						
Females vs. males	1.42 (0.99–2.18)	<0.048	1.28 (0.90–1.81)	0.167	1.41 (0.99–1.99)	0.049
**BAS**						
30–37 vs. ≥38	1.81 (1.16–2.80)	0.009	2.46 (1.67–3.64)	<0.001	1.60 (1.10–2.33)	0.015
≤29 vs. ≥38	1.80 (1.13–2.86)	0.014	3.42 (2.27–5.16)	<0.001	1.63 (1.10–2.42)	0.015
**BMI**						
≥25 vs. <25 kg/m^2^	1.05 (0.68–1.62)	0.825	0.49 (0.33–0.70)	<0.001	0.64 (0.44–0.91)	0.015

Abbreviations: BAS—body appreciation scale, BMI—body mass index, OR—odds ratio, CI—confidence intervals.

## Data Availability

The data presented in this study are available on request from the corresponding author.

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
