# Peer review of "Body Appreciation, Weight Status, and Weight Management Practices Among First-Year Students at Universities of Applied Sciences in Lithuania"

_medicina, 2025, doi:10.3390/medicina61071223_

Round 1
Reviewer 1 Report
Comments and Suggestions for Authors
Manuscript ID: medicina-3647869
Type of manuscript: Article
Title: Body Appreciation, Weight Status, and Weight Management Practices among First-Year Students at Universities of Applied Sciences in Lithuania
The paper titled "Body Appreciation, Weight Status, and Weight Management Practices among First-Year Students at Universities of Applied Sciences in Lithuania "is reasonably well-written and well-organized.
The authors have presented the study systematically.
The manuscript's global message is clear, and the information included can be a valuable consideration for many investigators.
There is only one comment that needs to be addressed by the authors. The manuscript presents the literature background with some gaps in the literature in the last 10 years.
I recommend the acceptance of this manuscript after taking into account my above comment.
Here's a concise response to each of the requested evaluation points:
1. What is the main question addressed by the research?
The main question addressed is: how are body appreciation, weight status (BMI), and weight management behaviors related among first-year university students in Lithuania, and how do these associations differ by gender?
2. Do you consider the topic original or relevant to the field? Does it address a specific gap in the field?
Yes, the topic is both original and relevant.
It addresses a specific gap by:
-focusing on positive body image (body appreciation) rather than the more frequently studied negative aspects (e.g., dissatisfaction, disordered eating).
-targeting a transitional population—first-year university students—who are particularly vulnerable to changes in self-image and behaviors.
-conducting research in Lithuania, where relatively little body image research exists compared to Western countries.
This regional and developmental focus brings fresh insight to a well-researched topic and highlights gender-specific vulnerabilities.
3. What does it add to the subject area compared with other published material?
The study adds:
(1)An examination of body appreciation as a protective factor, providing evidence that higher appreciation correlates with healthier behaviors and more accurate self-perception.
(2)A gender-differentiated analysis, showing that females report more dissatisfaction and weight control behaviors despite lower BMI.
(3)Data on harmful weight control methods (e.g., smoking), which enriches the literature on risky behaviors related to body image.
(4) A detailed look at motivating factors behind weight control, such as media influence and internalized beauty ideals.
These findings broaden the conversation beyond body dissatisfaction and toward preventative approaches centered on body positivity.
4. What specific improvements should the authors consider regarding the methodology?
While generally strong, the methodology could be improved by:
-mitigating self-report bias: BMI based on self-reported height and weight can be inaccurate. Incorporating objective measures or validation subsets would enhance accuracy.
-cross-sectional limitation: The design limits conclusions about causality. A longitudinal follow-up would better capture behavior and attitude changes over time.
-more detailed behavioral categories: Weight loss methods could be analyzed in separate categories (healthy vs. unhealthy) rather than as a single block.
-representation and recruitment: The authors can discuss possible selection bias (e.g., response rate, motivation to participate) and how it may affect generalizability.
5. Are the conclusions consistent with the evidence and arguments presented and do they address the main question posed?
Yes, the conclusions are consistent and well-supported.
They clearly tie back to the research question and are grounded in the results.
6. Are the references appropriate?
Yes, the references are appropriate. But the manuscript presents the literature background with some gaps in the literature in the last 10 years.
7. Any additional comments on the tables and figures?
No
Overall Assessment
This study is a well-executed, contextually valuable contribution to the field of body image research, particularly in transitional youth populations.
Author Response
Response to Reviewer 1 Comments
We would like to thank you for your efforts and for taking the time to read and revise our manuscript. We appreciate your comments and suggestions. We hope that we have successfully addressed all of the concerns raised, and we believe that the manuscript has been substantially improved. Our detailed responses to the comments and the description of the changes we have made to the manuscript are provided below.
Comment 1: The manuscript presents the literature background with some gaps in the literature in the last 10 years.
Response 1: Following the reviewer's recommendations, we replaced several older articles with newer ones (Ref. 12,13) and added some new articles (Ref. 32-34, 41). Most cited articles were published in the last five years.
Comment 2: BMI based on self-reported height and weight can be inaccurate. Incorporating objective measures or validation subsets would enhance accuracy.
Response 2: We agree with the limitation of self-reported height and weight data noticed by the reviewer. We mentioned this limitation of our study in the Discussion section: ‘The study relies on self-reported data for height and weight. Previous research has shown that self-reported anthropometric measurements are generally accurate for estimating BMI at the population level; however, there is still a risk of reporting bias [38]. Participants may underreport their weight or overestimate their height. University students might provide more accurate reports due to their higher education status and young age. In Lithuania, it is mandatory to take anthropometric measurements for every school child before the start of the school year. Thus, most young Lithuanians are aware of their height and weight. Additionally, our study was anonymous, which ensured confidentiality and encouraged participants to report accurate data. While most studies have found limited underestimation of self-reported BMI, which can affect prevalence estimates for BMI categories, the researchers agree that using self-reported BMI is appropriate for association analyses.’ (lines 373-384).
Comment 3: The design limits conclusions about causality. A longitudinal follow-up would better capture behavior and attitude changes over time.
Response 3: The limitation of cross-sectional study design is also mentioned in the Discussion section: ‘The primary limitation of this study is its cross-sectional design, which does not allow the establishment of causal relationships.’ (lines 372-373).
We also recommended conducting a prospective study in the future: ’Future studies should incorporate longitudinal designs and consider the role of additional psychosocial and environmental factors, including media exposure and mental health status, to better understand the dynamics of body image development during early adulthood.’ (lines 421-424).
Comment 4: Weight loss methods could be analyzed in separate categories (healthy vs. unhealthy) rather than as a single block.
Response 4: We did not categorize weight loss methods as healthy and unhealthy. Our goal was to explore how often different weight control methods are used by students. We did not ask about the diet and physical activity methods in detail, so we could not individualize the suitability of the methods. Using this classification within the stated limitations could lead to potential misunderstandings.
Comment 5: The authors can discuss possible selection bias (e.g., response rate, motivation to participate) and how it may affect generalizability.
Response 5: Thank you for raising this important issue. We recognize that selection bias may have affected our findings. We have elaborated on this limitation in the Discussion section: ‘Selection bias may be a concern since participation was voluntary, which could have attracted students particularly interested in health or body image issues. To enhance sample diversity, we included students from different faculties across four major universities of applied sciences and a variety of study programs. The overall response rate was modest, which is a common limitation of online surveys targeting students and limiting the generalizability of the study results. First-year students may differ from older students, which also limits the study results generalizability. Additionally, male participation was significantly lower, partly due to the gender distribution of students within the selected study programs. This discrepancy may affect the reliability of gender-based comparisons.’ (lines 386-395).

Reviewer 2 Report
Comments and Suggestions for Authors
dear authors
thank you for sending this paper to the journal, please take into consideration these points:
1-about sampling and randomization
"Selected faculties were randomly selected" is vague and redundant. Consider rewriting as:
"Faculties were randomly selected within each institution, and all enrolled first-year students in these faculties were invited to participate via institutional email."
2- participation details:
Mention response rate or how many were invited, if available.
Clarify inclusion/exclusion criteria (for example., only first-year students, any exclusions based on age or incomplete data?).
3-is there any bias in the reported BMI?
4-please add the validity of the questionnair, is it previousely used?
5-Define more clearly how you matched BMI categories with perception categories if possible
6- Lines 121–123 and 129 describe similar variables.
7-Specify whether participants could choose multiple methods
8-can you report the Cronbach’s alpha
9-is the alpha error 0.05? please state
10-why did you use tertiles? justification needed
11-did you assess multi collinearity?
12-in table 3 just mention the reference group, no need to repear OR =1
13-please provide the figure comparing the ORs
14-in the conclusion section please avoid repeating all results, only state the conclusion of your study
Best regards
Author Response
Response to Reviewer 2 Comments
We would like to thank you for your efforts and time to read and revise our manuscript. We appreciate your comments and suggestions. We hope that we have successfully addressed all of the concerns raised, and we believe that the manuscript has been substantially improved. Our detailed responses to the comments and the description of the changes we have made to the manuscript are provided below.
Comment 1: About sampling and randomization. "Selected faculties were randomly selected" is vague and redundant. Consider rewriting as: "Faculties were randomly selected within each institution, and all enrolled first-year students in these faculties were invited to participate via institutional email."
Response 1: Thank you for your comment. We have corrected the sentence as you suggested. (lines 92-94).
Comment 2: Mention response rate or how many were invited, if available
Response 2: We added the number of invited students in the Methods section: ‘Out of the 3253 students invited to participate in the study, a total of 721 students (221 males and 500 females) completed the self-administered questionnaire.’ (lines 99-100).
Comment 3: Clarify inclusion/exclusion criteria (for example., only first-year students, any exclusions based on age or incomplete data?).
Response 3: We clarified that only first-year students from selected faculties were invited to participate, without exclusions. Also, we stated that incomplete 12 questionnaires were excluded from analysis. ‘Out of the 3253 students invited to participate in the study, a total of 721 students (221 males and 500 females) completed the self-administered questionnaire. Twelve questionnaires were excluded from the analysis due to incomplete data, leaving 709 students' responses (216 males and 493 females) for analysis.’ (lines 89-102).
Comment 4: Is there any bias in the reported BMI?
Response 4: We agree with the limitation of self-reported height and weight data noticed by the reviewer. We mentioned this limitation of our study in the Discussion section: ‘The study relies on self-reported data for height and weight. Previous research has shown that self-reported anthropometric measurements are generally accurate for estimating BMI at the population level; however, there is still a risk of reporting bias [38]. Participants may underreport their weight or overestimate their height. University students might provide more accurate reports due to their higher education status and young age. In Lithuania, it is mandatory to take anthropometric measurements for every school child before the start of the school year. Thus, most young Lithuanians are aware of their height and weight. Additionally, our study was anonymous, which ensured confidentiality and encouraged participants to report accurate data. While most studies have found limited underestimation of self-reported BMI, which can affect prevalence estimates for BMI categories, the researchers agree that using self-reported BMI is appropriate for association analyses.’ (lines 373-384).
Comment 5: Please add the validity of the questionnaire, is it previously used?
Response 5: We included the following explanation in the Methods section: ‘A self-developed questionnaire was created for this study to evaluate actual and perceived weight status, body weight satisfaction, concerns about weight gain, efforts to lose weight, and factors that encourage weight loss. The questions were selected from validated questionnaires used in our prior student research [19].’ ‘The BAS has been previously validated among students in Lithuania, demonstrating that it is a reliable and valid tool for measuring body appreciation [20].’ (lines 109-112 and 146-147).
Comment 6: Lines 121–123 and 129 describe similar variables.
Response 6: Thank you for pointing out this mistake. The second sentence has been removed.
Comment 7: Specify whether participants could choose multiple methods.
Response 7: Yes, the question about weight-loss methods was a multiple-choice question. We explained it in the Methods section: ‘Students had the option to select multiple weight-loss methods.’ (lines 134-135).
Comment 8: Can you report the Cronbach’s alpha.
Response 8: Cronbach's alpha was included in the Statistical Analysis subsection: ‘To evaluate the internal consistency of the BAS-2 scale, we calculated Cronbach's alpha, which was found to be 0.962.’ (lines 157-158).
Comment 9: Is the alpha error 0.05? please state
Response 9: We stated in the Statistical Analysis subsection: ‘P values of less than 0.05 were considered to be statistically significant.’(lines 156-161).
Comment 10: Why did you use tertiles? justification needed
Response 10: We have included the requested justification in the Statistical Analysis subsection, as suggested by the reviewer: ‘As most of the analyzed variables were categorical, we divided the BAS-2 scores into tertiles using cutoff points of 29 and 37. The tertiles represent low, medium, and high levels of body appreciation, allowing us to apply statistical analyses suitable for categorical variables.’ (lines 156-161).
Comment 11: Did you assess multicollinearity?
Response 11: Multicollinearity was assessed prior to conducting the logistic regression analysis. For all models, the VIF values ranged from 1 to 2, indicating that the variables are not highly correlated.
Comment 12: in table 3 just mention the reference group, no need to repeat OR =1
Response 12: We corrected Table 3 and Table 4, as suggested by the reviewer. (lines 222 and 253).
Comment 13: Please provide the figure comparing the ORs
Response 13: We believe that the figure would replicate the results of the table, so we did not include the figure in the article.
Comment 14: In the conclusion section please avoid repeating all results, only state the conclusion of your study.
Response 14: The conclusions have been revised. (lines 413-424).

Reviewer 3 Report
Comments and Suggestions for Authors
- The description mentioned "random selection", but did not explain the specific method and process of randomization (for example, whether it was simple random sampling or stratified sampling). This may affect the reproducibility of the study and the representativeness of the results. Only students from Lithuanian University of Applied Sciences were included, and there are doubts whether the sample can represent the wider university student population. In addition, possible selection bias was not discussed, such as students who were only interested in weight management might be more inclined to participate in the survey.
- All data are based on self-reports, which may be subject to recall bias or social desirability bias. The article did not mention how to mitigate these biases, such as whether guidance instructions were provided to improve accuracy.
- Although the content of the questionnaire was mentioned, the detailed source of the questionnaire (except for the BAS-2 scale) or the validation process were not provided, and its content validity could not be evaluated.
- The study divided weight perception into three groups: "too thin", "just right", and "too fat", without further distinguishing subcategories such as "a little overweight" and "very overweight", which may mask subtle differences.
- Tertiles were used to group BAS-2 scores, but it was not explained why this method was chosen. Is there any theoretical or practical basis? In addition, the interpretation of the grouping and its impact on statistical analysis were not discussed.
- Likert scales are usually considered to be ordered variables, but it was not clearly stated whether they were treated as continuous variables or ordered categorical variables for statistical analysis, which may affect the assumptions and interpretations of the model.
- The discussion section mainly cited a large number of existing studies to support the results, but did not fully highlight the unique contributions of this study. For example, whether new gender difference patterns were found, or unique phenomena related to the specific sociocultural background of Lithuanian students were not clearly stated in the discussion.
- There is a lack of in-depth exploration of the potential mechanisms of some key findings. For example, why do students with high body appreciation not actively participate in weight management behaviors even if they are overweight? The psychological or sociocultural mechanisms behind this need further analysis.
- The sample of this study came from Lithuania, but the discussion did not conduct an in-depth analysis of the specific local sociocultural background (such as beauty standards and health concepts). Only international research results were cited, lacking an explanation of localized characteristics.
- Although the impact of social media on body image is mentioned, the discussion is relatively general and does not incorporate specific data or analysis (such as whether students influenced by social media show lower body appreciation or more unhealthy behaviors).
- The conclusion section makes a number of suggestions (such as "promote body appreciation" and "reduce the spread of social beauty ideals"), but these suggestions are relatively abstract and do not provide specific actionable measures. For example, how can these suggestions be implemented in university health promotion projects?
no
Author Response

(The authors gave the same response as above.)

Reviewer 4 Report
Comments and Suggestions for Authors
Comments to Authors
This study showed that: a) body appreciation is linked to healthier weight perceptions and behaviours; b) interventions that enhance body appreciation may help reduce body dissatisfaction and prevent unhealthy weight control practices, especially among female students.
Clinicians and researchers may struggle with appropriate terminology when discussing body size [1]. Pathologizing larger bodies has led to use of medicalized terms [1]. Previous studies have focused on terminology preferences among participants not in larger bodies, leaving out those most affected by the terminology. Fat-related and weight-neutral terms may be associated with more positive outcomes, challenging advocacy for person-first medicalized language [1]. An empirical approach to self-esteem and its associated factors is crucial during youth, when ranking and physical appearance significantly impact self-esteem [2]. Enhancing self-esteem helps students appreciate individual characteristics and maintain a positive body image despite unhealthy exposures [2].
Authors are kindly requested to emphasize the current concepts about these issues in the context of recent knowledge and the available literature. This articles should be quoted in the References list.
References
- Words are heavy: Weight-related terminology preferences are associated with larger-bodied people's health behaviors and beliefs. Body Image. Published online February 22, 2025. doi:10.1016/j.bodyim.2025.101860.
- Self-esteem, body image, and associated factors among female and male university students: A cross-sectional study. J Educ Health Promot. 2025; 14: 45. Published 2025 Feb 28. doi:10.4103/jehp.jehp_960_24.
Author Response
Response to Reviewer
We would like to thank you for your efforts and time to read and revise our manuscript. We appreciate your comments and suggestions. We hope that we have successfully addressed all of the concerns raised, and we believe that the manuscript has been substantially improved. Our detailed responses to the comments and the description of the changes we have made to the manuscript are provided below.
Comment 1: Clinicians and researchers may struggle with appropriate terminology when discussing body size [1]. Pathologizing larger bodies has led to use of medicalized terms [1]. Previous studies have focused on terminology preferences among participants not in larger bodies, leaving out those most affected by the terminology. Fat-related and weight-neutral terms may be associated with more positive outcomes, challenging advocacy for person-first medicalized language [1]. An empirical approach to self-esteem and its associated factors is crucial during youth, when ranking and physical appearance significantly impact self-esteem [2]. Enhancing self-esteem helps students appreciate individual characteristics and maintain a positive body image despite unhealthy exposures [2].
Authors are kindly requested to emphasize the current concepts about these issues in the context of recent knowledge and the available literature. These articles should be quoted in the References list.
Words are heavy: Weight-related terminology preferences are associated with larger-bodied people's health behaviors and beliefs. Body Image. Published online February 22, 2025. doi:10.1016/j.bodyim.2025.101860.
Self-esteem, body image, and associated factors among female and male university students: A cross-sectional study. J Educ Health Promot. 2025; 14: 45. Published 2025 Feb 28. doi:10.4103/jehp.jehp_960_24
Response 1:
We sincerely thank the reviewer for this insightful comment. We agree that the terminology used to describe body size and the importance of self-esteem during youth are both critical factors influencing body image and related behaviours. In response, we integrated your recommended recent findings into the Discussion section of the manuscript. Specifically, we added a statement acknowledging that weight-related terminology can influence emotional and behavioural responses, especially among individuals with larger bodies. We cited Robbins et al. (2025), who found that weight-neutral and fat-related terms are often perceived more positively than medicalized language and may reduce stigma and support health-promoting behaviours. This update is now included in the paragraph discussing the impact of obesity stigma and societal attitudes: ‘The language used to describe body size can significantly impact people's attitudes. Research indicates that using weight-neutral and fat-related terms tends to carry less stigma and encourages more supportive health behaviours, particularly for individuals with larger bodies, compared to medicalized language (Robbins et al., 2025). (lines 330-334).
Additionally, we addressed the importance of self-esteem in shaping body image during early adulthood by citing the recent study by Achak et al. (2025). We expanded the discussion of how self-esteem may act as a protective factor against internalized appearance pressures and negative body image outcomes. This addition supports our findings related to body appreciation and offers further context for the need to strengthen self-esteem in student health promotion programs. ‘Additionally, recent research emphasizes the importance of self-esteem in shaping body image among university students, especially during early adulthood when concerns about social status and appearance are more pronounced. Findings indicate that lower self-esteem is associated with a higher motivation to change one's weight, particularly among females. Improving self-esteem has been shown to foster body appreciation and decrease vulnerability to internalized stigma.’(Achak et al., 2025) (lines 295-300).

Reviewer 5 Report
Comments and Suggestions for Authors
While the manuscript offers valuable insights into the associations between body appreciation, weight status, and weight management behaviors among first-year university students in Lithuania, several limitations should be addressed or acknowledged to strengthen the interpretation and generalizability of the findings.
Although admitted at the end of the discussion, most of these caveats ought to be carefully considered and adequately addressed.
- Key variables, including height, weight, and weight management practices, are based on self-reported data. Such measures are prone to social desirability and recall biases, which may result in inaccurate BMI calculations or underreporting of unhealthy behaviors such as smoking or extreme dieting.
-
The sample is drawn from first-year students attending four universities of applied sciences in Lithuania. While the sample size is adequate, the findings may not generalize to:
-
Students in research universities or different academic disciplines
-
Older students or those in later years of study
-
Populations in different sociocultural or national contexts
The authors may consider discussing how institutional and cultural contexts may influence the applicability of the results elsewhere.The study is situated within a specific cultural setting, where societal beauty norms and body image pressures may differ from other regions. Greater discussion of cultural influences on body appreciation and how these may shape the findings would enhance the reader’s understanding and appreciation of context.
-
- The sample is disproportionately female (69.5%), which may limit the generalizability of gender comparisons. The statistical power to detect differences among male participants may be insufficient, and this imbalance should be acknowledged as a potential source of bias in sex-stratified analyses.
- The regression models do not appear to control for potential confounders such as socioeconomic status, mental health (e.g., depression, anxiety), or social media use—all of which are known to influence both body appreciation and weight-related behaviors. This may inflate or obscure observed associations.
- Some behaviors (e.g., smoking as a weight-control strategy) may oversimplify the multifaceted nature of disordered eating and compensatory behaviors. A more nuanced behavioral categorization could improve the interpretability of the data.
The manuscript would benefit from a more comprehensive limitations section that addresses the above concerns. Additionally, expanding on potential implications for public health interventions, particularly how culturally tailored strategies might improve body appreciation and prevent maladaptive behaviors, would enhance the relevance and impact of the study.
Author Response
Response to Reviewer
We would like to thank you for your efforts and time to read and revise our manuscript. We appreciate your comments and suggestions. We hope that we have successfully addressed all of the concerns raised, and we believe that the manuscript has been substantially improved. Our detailed responses to the comments and the description of the changes we have made to the manuscript are provided below.
Comment 1: Key variables, including height, weight, and weight management practices, are based on self-reported data. Such measures are prone to social desirability and recall biases, which may result in inaccurate BMI calculations or underreporting of unhealthy behaviors such as smoking or extreme dieting.
Response 1: We included this issue in study limitations: ‘The study relies on self-reported data for height and weight. Previous research has shown that self-reported anthropometric measurements are generally accurate for estimating BMI at the population level; however, there is still a risk of reporting bias [45]. Participants may underreport their weight or overestimate their height. University students might provide more accurate reports due to their higher education status and young age. In Lithuania, it is mandatory to take anthropometric measurements for every school child before the start of the school year. Thus, most young Lithuanians are aware of their height and weight. Additionally, our study was anonymous, which ensured confidentiality and encouraged participants to report accurate data. While most studies have found limited underestimation of self-reported BMI, which can affect prevalence estimates for BMI categories, the researchers agree that using self-reported BMI is appropriate for association analyses.’ (lines 373-384).
Comment 2: The sample is drawn from first-year students attending four universities of applied sciences in Lithuania. While the sample size is adequate, the findings may not generalize to: Students in research universities or different academic disciplines, Older students or those in later years of study, Populations in different sociocultural or national contexts.
Response 2: We have discussed this issue: ‘To enhance sample diversity, we included students from different faculties across four major universities of applied sciences and a variety of study programs. The overall response rate was modest, which is a common limitation of online surveys targeting students and limiting the generalizability of the study results. First-year students may differ from older students, which also limits the study results generalizability.’ (lines 388-392).
Comment 3: The authors may consider discussing how institutional and cultural contexts may influence the applicability of the results elsewhere. The study is situated within a specific cultural setting, where societal beauty norms and body image pressures may differ from other regions. Greater discussion of cultural influences on body appreciation and how these may shape the findings would enhance the reader’s understanding and appreciation of context.
Response 3: Following reviewer’s comment, we discussed cultural influences on body appreciation and the applicability of the results elsewhere.
‘Our findings are consistent with national data indicating that Lithuanian female students have a more accurate perception of their weight status compared to males, but they express greater dissatisfaction [20]. Longitudinal data from studies conducted among university students in Kaunas between 2000 and 2017 indicated that perceptions of body image and weight control behaviors remained stable over two decades. Notably, it was subjective perceptions, rather than objective weight status, that influenced weight control behaviours [23]. This paradox, where a more accurate perception of body image aligns with greater dissatisfaction, has not been extensively documented in Lithuanian literature. It may suggest that young Lithuanian women are internalizing Western ideals of thinness.
Our results highlight gender-specific patterns and sociocultural influences. Female students not only reported more dissatisfaction and weight-related behaviours, but also greater media-driven pressure to conform to thin ideals. In contrast, males were more likely to perceive themselves as too thin and showed less concern with weight gain. These patterns, also seen in previous Lithuanian studies, suggest that Western beauty norms are integrated into local contexts shaped by societal transformation [42].’
‘To address body image concerns among university students, it is crucial to promote accurate weight perception, strengthen media literacy, and provide access to gender-sensitive psychological support. Body positivity campaigns and media literacy interventions have been shown to reduce internalization of harmful beauty ideals and improve body image outcomes [9,44]. In Lithuania, most universities have already taken practical steps to address student well-being. For example, many institutions offer free, anonymous psychological consultations to address stress and mental health concerns. A unique example is the Lithuanian University of Health Sciences, which provides lifestyle medicine consultations for students focused on individualized strategies in physical activity, nutrition, stress management, and sleep. These services, offered free of charge, help students develop healthier lifestyle habits that can support a more positive body image and reduce reliance on appearance-based weight control. Integrating such services into broader university health promotion strategies may serve as a valuable model for enhancing both physical and emotional well-being in Lithuanian and international academic settings.’ (lines 338-366).
Comment 4: The sample is disproportionately female (69.5%), which may limit the generalizability of gender comparisons. The statistical power to detect differences among male participants may be insufficient, and this imbalance should be acknowledged as a potential source of bias in sex-stratified analyses.
Response 4: We have included the following limitations of the study: ‘Additionally, male participation was significantly lower, partly due to the gender distribution of students within the selected study programs. This discrepancy may affect the reliability of gender-based comparisons.’ (lines 392-395).
Comment 5: The regression models do not appear to control for potential confounders such as socioeconomic status, mental health (e.g., depression, anxiety), or social media use—all of which are known to influence both body appreciation and weight-related behaviors. This may inflate or obscure observed associations.
Response 5: Those study limitations were also mentioned: ‘Important data on several potentially relevant factors, such as socioeconomic status, psychological well-being, and media usage patterns, were not collected. These contextual factors could influence body image and weight management behaviors, as well as the analyzed associations. Future research should take these factors into consideration.’ (lines 396-400).
Comment 6: Some behaviors (e.g., smoking as a weight-control strategy) may oversimplify the multifaceted nature of disordered eating and compensatory behaviors. A more nuanced behavioral categorization could improve the interpretability of the data.
Response 6: The methods for controlling body weight among students were chosen based on previous research data, identifying the most commonly used techniques. We acknowledge your observation that there are more unhealthy methods for regulating body weight, and we will consider these in our future research. We included this limitation: ‘Weight control methods listed in the questionnaire do not include all the techniques that students might use.’(lines 395-396).
Comment 7: The manuscript would benefit from a more comprehensive limitations section that addresses the above concerns. Additionally, expanding on potential implications for public health interventions, particularly how culturally tailored strategies might improve body appreciation and prevent maladaptive behaviors, would enhance the relevance and impact of the study.
Response 7: The limitations of the study have been revised to incorporate your comments. (lines 373-396).

Reviewer 6 Report
Comments and Suggestions for Authors
This work may be more appropriate for a journal with a brother or more introductory scope
Author Response
Response to Reviewer. Comments.
We would like to thank the reviewer for the efforts and time to read our manuscript. We appreciate the reviewer’s comment regarding the scope and journal suitability of our article – ‘this work may be more appropriate for a journal with a brother or more introductory scope’.
While we understand the suggestion that the study may align with journals of a broader or more introductory nature, we believe that our work offers meaningful contributions to the field of public health and epidemiology. Our manuscript addresses key aims of the Medicina journal's Epidemiology & Public Health subsection. Furthermore, the study aligns closely with the scope of the special issue “Advances in Health, Lifestyle and Environmental Risk Factors Monitoring,” as it investigates the associations between body appreciation, body weight status, body weight perception, and weight control behaviours among university students—a population increasingly recognized as at-risk for disordered eating and mental health challenges.
Importantly, our findings contribute to the underrepresented Eastern European context, offering region-specific evidence that can inform local and international public health strategies focused on youth well-being. We also employ a validated psychometric tool (BAS-2), multivariable regression analysis, and a population-based sample that strengthens the study’s applicability for health promotion and policy development.
For these reasons, we hope the revised manuscript is now more clearly positioned within the journal’s thematic scope. We thank the reviewer again for their feedback and consideration.

Round 2
Reviewer 2 Report
Comments and Suggestions for Authors
dear authors
thank you for explanations, please note the ollowings:
1- response rate, as percentage would be more informative
2-about Questionnaire validity, which items were adapted?
3-further analysis and clarification using the continuous BAS-2 score if appropriate
4-please specify if VIF applied to all regression models or not?
5-the conclusion must be revised, currently is generic and overly broad, better link findings to practical applications are suggested
Author Response
We would like to thank you for your efforts and time to read and revise our manuscript. We appreciate your comments and suggestions. We hope that we have successfully addressed all of the concerns raised, and we believe that the manuscript has been substantially improved. Our detailed responses to the comments and the description of the changes we have made to the manuscript are provided below.
Comment 1: response rate, as percentage would be more informative.
Response 1: Thank you for your comment. The relevant sentence in the Materials and Methods section has been updated:
‘Out of the 3253 students invited to participate in the study, 721 (221 males and 500 females) completed the self-administered questionnaire (response rate 22.2%). (lines 99-100).
Comment 2: about Questionnaire validity, which items were adapted?
Response 2: The following questions were taken from previously validated international student surveys:
- In your opinion, are you:
Far too thin / A little too thin / Just right / A little overweight / Very overweight
- How satisfied are you with your current weight?
Very satisfied / Somewhat satisfied / Somewhat dissatisfied / Very dissatisfied
- Would you worry if you gained weight?
Never / Very rarely / Sometimes often / Very often
We explained it in the Methods section. (lines 115-128).
We included the following explanation in the Methods section: ‘The questions about weight perception, satisfaction with weight, and concerns about weight gain were selected from validated questionnaires used in our prior student research’ [19]. (lines 109-110).
Comment 3: further analysis and clarification using the continuous BAS-2 score if appropriate.
Response 3: We performed additional analyses using the continuous BAS-2 score. Spearman correlation coefficients were calculated to examine associations between the BAS-2 score and selected variables among students with normal weight and those with overweight (BMI ≥ 25) (Table 1).
Table 1. Spearman correlation coefficients for associations between the BAS-2 score and selected variables.
Variables |
BMI<25 |
BMI≥25 |
||
correlation coefficients |
p-Value |
correlation coefficients |
p-Value |
|
Perceived weight status |
-0.143 |
0.001 |
-0.467 |
<0.001 |
Accuracy of weight perception |
0.181 |
<0.001 |
-0.466 |
<0.001 |
Satisfaction with body weight |
0.271 |
<0.001 |
0.149 |
0.039 |
Worries about gaining weight |
-0.261 |
<0.001 |
-0.551 |
<0.001 |
Trying to lose weight |
-0.019 |
0.013 |
-0.245 |
0.001 |
These findings are consistent with the associations reported in Table 2 of the manuscript. However, we believe Table 2 offers a clearer overview of the relationships between the BAS-2 score and the selected variables, beyond simple correlation coefficients.
We also attempted regression analyses using the continuous BAS-2 score. However, its distribution was non-normal, which limited the suitability of standard logistic regression. We performed logistic regression analyses, using the BAS-2 score as an independent variable to predict various weight management methods (dependent variables).
The only statistically significant association observed was with smoking, where each one-point increase in the BAS-2 score was associated with a lower odds of smoking (OR = 0.903; 95% CI: 0.872–0.935).
Comment 4: please specify if VIF applied to all regression models or not?
Response 4: Yes, VIF was calculated for every logistic regression model, and it was between 1 and 2.
Comment 5: the conclusion must be revised, currently is generic and overly broad, better link findings to practical applications are suggested
Response 5: Thank you for your comment. We have revised the conclusion section to avoid repetition of results, as you mentioned in the first review, and to ensure it is more practically oriented.
‘This study demonstrated that body appreciation is associated with more accurate weight perception, greater satisfaction with body weight, and healthier weight management behaviours among university students. Strengthening body appreciation, particularly among female students, may help to reduce dissatisfaction with body weight and prevent engagement in harmful weight control practices. University health promotion programmes should incorporate strategies aimed at enhancing body appreciation, promoting media literacy, providing psychological support, and encouraging healthy lifestyle habits to support realistic body image and health-oriented behaviours during the transition to adulthood’. (lines 412-420).

Reviewer 5 Report
Comments and Suggestions for Authors
Thank you for having attempted to address all the raised issues. The new version of the manuscript reads well and, in fact, was substantially improved.
Author Response
Thank you for your positive feedback. We appreciate your comments and the opportunity to improve the manuscript.